# CS-SHAPLEY: Class-wise Shapley Values for Data Valuation in Classification

**Stephanie Schoch**[*]    **Haifeng Xu**[†]    **Yangfeng Ji**[*]
[*]Department of Computer Science, University of Virginia, Charlottesville, VA 22904
[†]Department of Computer Science, University of Chicago, Chicago, IL 60637

## Abstract

Data valuation, or the valuation of individual datum contributions, has seen growing interest in machine learning due to its demonstrable efficacy for tasks such as noisy label detection. In particular, due to the desirable axiomatic properties, several Shapley value approximation methods have been proposed. In these methods, the value function is typically defined as the predictive accuracy over the entire development set. However, this limits the ability to differentiate between training instances that are helpful or harmful to their own classes. Intuitively, instances that harm their own classes may be noisy or mislabeled and should receive a lower valuation than helpful instances. In this work, we propose CS-SHAPLEY, a Shapley value with a new value function that discriminates between training instances' in-class and out-of-class contributions. Our theoretical analysis shows the proposed value function is (essentially) the unique function that satisfies two desirable properties for evaluating data values in classification. Further, our experiments on two benchmark evaluation tasks (data removal and noisy label detection) and four classifiers demonstrate the effectiveness of CS-SHAPLEY over existing methods. Lastly, we evaluate the "transferability" of data values estimated from one classifier to others, and our results suggest Shapley-based data valuation is transferable for application across different models.

## 1   Introduction

Data valuation methods aim to quantify the contribution of each datum to the predictive performance of a learning model. Among these, Shapley values have been proposed as a means to identify helpful or harmful data [3, 10]. A number of approximations and extensions for Shapley-based data valuation have been developed, with demonstrable efficacy for tasks such as mislabeled or noisy example detection and data selection [3, 10, 14, 4, 11]. The performance gains of Shapley-based approaches over alternative data valuation methods have typically been attributed to the axiomatic basis of Shapley values that satisfies fairness guarantees from cooperative game theory. Importantly, Shapley values rest on an underlying assumption that a game is well-represented by its value function [21].

The value function of prior Shapley-based data valuation methods has typically been defined as the predictive accuracy over the entire development set. However, in the context of valuing data for learning models on classification tasks, this may have limited ability to differentiate helpful or harmful training instances. Consider the case where we want to evaluate the value of data points $i$ and $j$ for a binary classification task, where both points belong to class 1. As shown in the real world example provided in Figure 1, if the predictive accuracy on the development set is the same when adding each point individually, then the contribution of these two data points is considered to be equivalent. However, *how* $i$ and $j$ contribute to the classifier differs. To be specific, the contribution of data point $i$ to class 1 is positive (helpful), while the contribution of $j$ to class 1 is negative (harmful). Similar distinction between training instances that are "helpful" and "harmful" to their own class

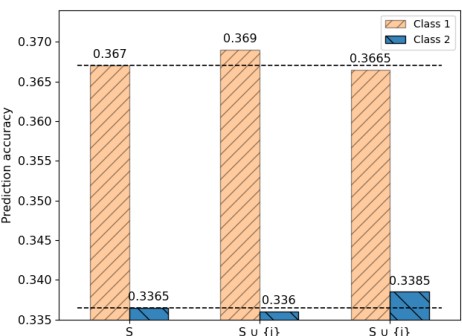

Figure 1: Development accuracy by class when adding two different points, $i$ and $j$, to the training set of a binarized version of CIFAR10, using logistic regression (the experiment setup is provided in section 5). Both points belong to class 1 and produce the same overall development accuracy change. However, $i$ increases the in-class accuracy, and $j$ decreases the in-class accuracy. If measuring contribution using the overall predictive accuracy, $i$ and $j$ will have equivalent contributions. In contrast, by differentiating between in-class and out-of-class accuracy changes, the proposed value function considers $i$ to have a larger contribution than $j$.

has previously been used for post-hoc analysis in prior data contribution literature, such as influence functions [12, 20].

In this work, we propose a class-wise value function that differentiates between the contribution of a data point to its own class and to other classes. Consider the running example in Figure 1, $i$ increases in-class accuracy, while $j$ decreases in-class accuracy. Intuitively, $i$ should receive a higher value than $j$, as $j$ could be a mislabeled, adversarial, or otherwise noisy instance. Our proposed class-wise value function $v_y(S \cup \{i\})$ measures the contribution of data point $i$ based on its class label $y = 1$, where the accuracy of class 1 is a measure of contribution of $i$ and the accuracy of class 2 is a weighting factor. The definition of this new value function is detailed in section 3. For the example in Figure 1, this new class-wise value function measures the contribution as $v_1(S \cup \{i\}) > v_1(S \cup \{j\})$. A key conceptual message of this paper is to demonstrate that such distinction of in-class and out-of-class accuracy not only leads to desirable theoretical properties for measuring data values in classification (section 4) but also exhibits high efficacy in extensive empirical evaluations (section 5).

**Contributions.** 1) we propose a new value function that differentiates between in-class and out-of-class contribution for computing Shapley values on classification datasets; 2) we theoretically show that this value function is essentially the unique choice — up to some freedom to change a constant — that satisfies two desirable properties for data valuation in classification; 3) we perform a systematic evaluation on two benchmark tasks using four classifiers, nine datasets, and three baseline methods. Our results demonstrate that our method outperforms existing methods across almost all experimental conditions; 4) last but not least, we also propose a new evaluation task to measure the *transferability* of data values estimated from different classifiers; using the proposed transferability task, we show that Shapley-based data value estimates can be transferred across classifiers, including transfer to neural models.[1]

## 2   Related work

**Data valuation methods.** Shapley values are a foundational concept in cooperative game theory that ensures fair division of rewards in cooperative games [21]. In a machine learning setting, Shapley values have been applied to data valuation, i.e. quantifying the contribution of individual datum [3, 10]. Exact computation of Shapley-based data values, however, requires exhaustively retraining and evaluating marginal contributions of every datum using every possible data subset. To circumvent this, Shapley-based data values have been approximated with methods such as truncated Monte-Carlo Sampling [3], influence-based approximations of parameters changes [10], and federated learning [24]. To our knowledge, our work is the first to consider Shapley values induced by a value function that discriminates between in-class and out-of-class accuracy. In section 4, we theoretically analyze the desirable properties of class-wise Shapley values within the context of classification.

Other work that builds upon Shapley-based data values includes using the context of the underlying data distribution to increase valuation stability [4, 15], relaxing the Shapley efficiency axiom to reduce noise [14], and using $k$-nearest neighbor classifiers over pretrained feature embeddings as surrogates for larger models [11]. Notably, there are alternative methods to measure data contribution such as

---

[1]Code is available at `https://github.com/stephanieschoch/cs-shapley`

the leave-one-out method [2], influence functions [12], and reinforcement learning [26], however, these methods have not been proven to share the fairness guarantees of Shapley values.

**Applications of Shapley-based data values.** Prior work has demonstrated the benefits of using Shapley-based data values in many applications, such as mislabeled example detection [23, 3, 14], data selection for transfer learning [18] and active learning [5], and data sharing [22, 7]. The core idea behind these applications is that the Shapley value of a training instance indicates its contribution to a trained predictive model. By designing a new value function, our method aims to provide more effective estimates of data values and has the potential to apply to all of these applications. For real-world applications, we recognize the computational challenge of estimating Shapley values directly from classifiers used in practice (e.g., neural network models). Therefore, we also propose to systematically study the transferability of Shapley-based data values across different classifiers, in addition to evaluating on two benchmark evaluation tasks.

# 3 Proposed method: CS-SHAPLEY

## 3.1 Preliminaries

Consider a training dataset $T = \{(x_i, y_i)\}_{i=1}^n$ that contains $n$ training instances. Let $\mathcal{A}$ denote a classification algorithm and $v(S) : 2^T \to \mathbb{R}$ be a value function that evaluates the value of any subset of data $S \subseteq T$. For classification tasks, $v(\cdot)$ is often considered to be the classification accuracy on a development set $D$ [3, 14, 10, 11], and $v(S)$ represents the development accuracy $a_S(D)$ when the classifier is trained on $S$ and evaluated on $D$. For each data point $i$ in the training set, the Shapley value $\phi_i(T, \mathcal{A}, v)$ is defined as the average marginal contribution of $i$ to every possible subset $S \subseteq T \backslash \{i\}$:

**Definition 1** (Data Shapley value [21, 3])**.** *Given a value function $v(\cdot)$, the Shapley value $\phi_i(T, \mathcal{A}, v)$ for any data point $i$ is defined as*

$$\phi_i(T, \mathcal{A}, v) = \sum_{S \subseteq T \backslash \{i\}} \frac{v(S \cup \{i\}) - v(S)}{\binom{n-1}{|S|}} \qquad (1)$$

When the dataset $T$, classification model $\mathcal{A}$, and value function $v$ are clear from the context, we simply use $\phi_i$ to denote the Shapley value. Shapley values satisfy the following axioms [21]:

- Symmetry: if for all $S \subseteq T \backslash \{i, j\}, v(S \cup \{i\}) = v(S \cup \{j\})$, then $\phi_i = \phi_j$.
- Linearity: $\phi_i(v + w) = \phi_i(v) + \phi_i(w)$ for value functions $v$ and $w$.
- Null player: if for all $S \subseteq T \backslash \{i\}, v(S) = v(S \cup \{i\})$, then $\phi_i = 0$.
- Efficiency: $v(T) = \sum_{i \in T} \phi_i$.

Prior work usually considers $v(\cdot)$ to be the predictive accuracy on the development set. Recalling the example in Figure 1, this may not be an ideal setting to discriminate between harmful (or noisy) and helpful instances. Notably, this limitation cannot be addressed simply by switching to another development set level metric such as F1, precision, or recall; we will further illustrate this with an example in Appendix B. This key drawback motivates the development of a new value function, described in the following section, which has been designed to better differentiate between harmful and helpful instances.

## 3.2 Class-wise data Shapley

Along the previous lines of discussion, we suggest data for classification may contain implicit, pre-existing coalitions based on class membership, which should be accounted for when evaluating contributions. Motivated by this intuition, we propose a new value function that differentiates between the contribution of adding one instance to its own class vs. to other classes. The key idea behind our design is to use in-class accuracy as the measurement of contribution and out-of-class accuracy as a discounting factor. In this way, we gain the benefits of evaluating value on the class level, yet assure we do not assign high value to instances that may be detrimental to the out-of-class performance.

**Class-wise value function.** Consider the problem of estimating the contribution of a data point $i$, $(x_i, y_i)$, given a subset of training instances $S \subseteq T \backslash \{i\}$, and a development set $D$. To define a class-wise value function, we need to partition $D$ into two subsets $D_{y_i}$ and $D_{-y_i}$. $D_{y_i}$ contains the

development instances with the class label $y_i$ and $D_{-y_i}$ contains the development instances with the other labels. For multi-class classification, $D_{-y_i}$ has all the instances with labels other than $y_i$. Similarly, we have $S_{y_i}$ and $S_{-y_i}$ with $S = S_{y_i} \cup S_{-y_i}$. To measure the contribution of data point $i$ to its own class $y_i$ and to the other classes $-y_i$, we define two separate accuracy numbers, in-class accuracy $a_S(D_{y_i})$ and out-of-class accuracy $a_S(D_{-y_i})$, as the following

$$a_S(D_{y_i}) = \frac{\text{\# of correct predictions in } D_{y_i}}{|D|}, \quad a_S(D_{-y_i}) = \frac{\text{\# of correct predictions in } D_{-y_i}}{|D|} \quad (2)$$

Note that since $a_S(D_{y_i})$ and $a_S(D_{-y_i})$ share the same denominator, we have $a_S(D_{y_i}) + a_S(D_{-y_i}) = a_S(D)$, which is the accuracy on the whole development set. With $a_S(D_{y_i})$ and $a_S(D_{-y_i})$, our class-wise value function is defined as

$$v_{y_i}(S_{y_i}|S_{-y_i}) = a_S(D_{y_i}) \cdot e^{a_S(D_{-y_i})} \quad (3)$$

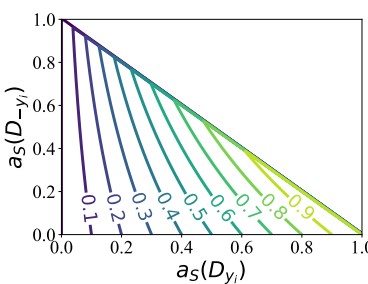

Figure 2: Contour plot of $v_{y_i}(S)$.

Figure 2 visualizes the contour plot of $v_{y_i}(S)$ based on different $a_S(D_{y_i})$ and $a_S(D_{-y_i})$. Between the two variables used in the value function, the significant factor is the in-class accuracy $a_S(D_{y_i})$. The effect of the out-of-class accuracy $a_S(D_{-y_i})$ is controlled by the value of $a_S(D_{y_i})$. Particularly, when $a_S(D_{y_i})$ is small, the effect of $a_S(D_{-y_i})$ can be ignored. To better understand how this value function works, assume $a_S(D_{y_i}) = 0.1$, which indicates class $y_i$ is difficult to learn. Under this condition, the value of adding an instance in this class is primarily from the prediction performance improvement of its own class, rather than that of other classes. This is a desirable property of the class-wise value function, which will be formally defined in section 4.

**Class-wise Shapley values.** With the new value function, the **C**lass-wi**S**e Shapley (CS-SHAPLEY) value of instance $i$ conditioned on any out-of-class "environment" $S_{-y_i}$ is defined as

$$\phi_i|S_{-y_i} = \sum_{S_{y_i} \subseteq T_{y_i} \setminus \{i\}} \frac{v_{y_i}(S_{y_i} \cup \{i\}|S_{-y_i}) - v_{y_i}(S_{y_i}|S_{-y_i})}{\binom{n-1}{|S_{y_i}|}}. \quad (4)$$

To compute the marginal CS-SHAPLEY value of instance $i$, we then simply average over all possible environmental data $S_{-y_i} \subseteq T_{-y_i}$ with equal weight, which leads to our following definition of the *Canonical* CS-SHAPLEY

$$\phi_i = \frac{1}{2^{|T_{-y_i}|}} \sum_{S_{-y_i}} [\phi_i|S_{-y_i}] \quad (5)$$

We remark that the word "canonical" here refers to our simple choice of equal weight $\frac{1}{2^{|T_{-y_i}|}}$ for each sampled out-of-class environment $S_{-y_i i} \subseteq T_{-y_i}$. More generally, one could possibly consider non-canonical and more sophisticated weights, e.g. weights depending on the size of $S_{-y_i}$. However, it turns out that the canonical choice in Equation (5) already performs very well in our experiments. Following the principle of Occam's Razor, we thus will stick with this canonical form for the remainder of this paper.

**Algorithm.** Exactly computing $\phi_i$ in Equation (5) requires averaging over exponentially many $S_{-y_i}$, which is computationally prohibitive. Thus we use a relatively small number of subsets $S_{-y_i} \subseteq T_{-y_i}$ for approximating $\phi_i$

$$\phi_i \approx \frac{1}{K} \sum_{S_{-y_i}^{(k)} \subseteq T_{-y_i}; k \in \{1,..,K\}} [\phi_i|S_{-y_i}^{(k)}]. \quad (6)$$

In our implementation, we use $K = 500$. Such approximation via samples is widely used in previous works [14], and has been proved to give good approximations under structural assumptions about the value function [17, 1]. Although the description above only talks about a single instance, the actual implementation of the algorithm is much more efficient, if we compute the values per class. The detailed implementation of our algorithm can be found in the pseudo-code deferred to Appendix A. At a high level, for any given class label $y$, the algorithm first samples a subset $S_{-y}$ from $T_{-y}$. Then,

for all the examples in class $y$, we adopt the truncated Monte Carlo algorithm [3] to estimate the conditional class-wise Shapley values defined in Equation (4). By repeating this procedure $K$ times, the CS-SHAPLEY value estimation is done by Equation (6). Before switching to another class, we normalize the estimated Shapley values by the in-class accuracy when using the whole training set to satisfy the efficiency axiom.

## 4   Theoretical justifications of the value function choice

In this section, we carry out a theoretical analysis to provide insight and justifications about our approach. We will formally prove that, to fulfil some desirable properties of a class-wise value function, the form that we adopt in Equation (3) is essentially the unique choice, up to the choice of the basis of the exponential function.

To distinguish the accuracy from the in-class and out-of-class development set, we start by assuming that the value function is *separable* and has the following generic form for any subset of data $S \subseteq T$:

$$v_{y_i}(S) = f(a_S(D_{y_i})) \cdot g(a_S(D_{-y_i})) \tag{7}$$

where $f, g$ are naturally assumed to be *continuous* and *monotone increasing* functions. For normalization reasons, without loss of generality, we further assume $f(0)g(0) = 0$. Next, we describe two additional desirable properties of the value function on any development set $D$:

- **Property 1: Priority of In-class Accuracy** (i.e., $a_S(D_{y_i})$). Specifically, for any $a_S(D_{y_i}) > 0$, we have $f(a_S(D_{y_i}))g(0) > f(0)g(1)$.
- **Property 2: In-class Value Additivity and Out-of-class Weight Discounting**. Specifically, for any *partitions* of in-class development set $D_{y_i} = D_{y_i,1} \cup D_{y_i,2}$ and out-of-class development $D_{-y_i} = D_{-y_i,1} \cup D_{-y_i,2}$, we have

$$
\begin{aligned}
f(a_S(D_{y_i})) \cdot g(a_S(D_{-y_i})) &= f(a_S(D_{y_i,1})) \cdot g(a_S(D_{-y_i,1})) \cdot g(a_S(D_{-y_i,2})) \\
&\quad + f(a_S(D_{y_i,2})) \cdot g(a_S(D_{-y_i,1})) \cdot g(a_S(D_{-y_i,2}))
\end{aligned}
\tag{8}
$$

The first property above tries to formalize the intuition that in-class accuracy should be prioritized. Concretely, the value function for getting positive in-class accuracy $a_S(D_{y_i})$ and 0 out-of-class accuracy is no less than getting even perfect out-of-class accuracy but 0 in-class accuracy. The following theorem shows that this property is the underlying reason of the observed contour line in Figure 2. This also justifies the adoption of Property 1.

**Theorem 1.** *Suppose the value function defined in Equation* (7) *satisfies the property of* Priority of In-class Accuracy, *then no contour lines will intersect the axis of* $a_S(D_{-y_i})$, *except the special line for* $f(a_S(D_{y_i})) \cdot g(a_S(D_{-y_i})) = 0$.[2]

*Proof of Theorem 1.* By the property of *Priority of In-class Accuracy*, we know $f(a_S(D_{y_i}))g(0) > f(0)g(1)$ for any $a_S(D_{y_i}) > 0$. By taking the limit of letting $a_S(D_{y_i}) \to 0$ in the above inequality, we have $\lim_{a_S(D_{y_i}) \to 0} f(a_S(D_{y_i}))g(0) = f(0)g(0) = 0 \geq f(0)g(1)$.

Next, we prove the theorem by contradiction. Suppose, for the purpose of contradiction, that there exists a $c > 0$ such that its contour line intersects the axis of $a_S(D_{-y_i})$ at some point $(0, y)$ for some $y \leq 1$. Then we have $f(0)g(y) = c$ by the definition of contour line. This however yields the following contradicting inequalities:

$$0 < c = f(0)g(y) \leq f(0)g(1) \leq 0 \tag{9}$$

where the last inequality is proved at the beginning of this proof. Therefore, it must be the case that the only countour line that can intersect the $a_S(D_{-y_i})$ is the $f(a_S(D_{y_i})) \cdot g(a_S(D_{-y_i})) = 0$ line. This concludes our proof. $\square$

The intuition behind the second property is based on the role of $f$ and $g$ in the definition. As a value measurement on the target class, $f(a_S(D_{y_i}))$ is expected to be the sum of the value of any two non-overlapped splits of $D_{y_i}$. In addition, as a weighting function $g$, the effect of $a_S(D_{-y_i})$ should be equivalent to applying the weights from $a_S(D_{-y_i,1})$ and $a_S(D_{-y_i,2})$ separately.

---

[2]Figure 2 is an example of such contour lines.

Theorem 2 shows that our previously defined value function $v_{y_i}(S) = v_{y_i}(S_{y_i}|S_{-y_i}) = a_S(D_{y_i}) \cdot e^{a_S(D_{-y_i})}$ is (essentially) the only choice that satisfies the two desirable properties above. This theoretically justifies our choice of the value function.

**Theorem 2.** *If the value function satisfies both Property 1 and 2 above, then it must have the form* $v_{y_i}(S) = c' a_S(D_{y_i}) \cdot c^{a_S(D_{-y_i})}$ *for some constant $c > 1, c' > 0$.[3]*

**Remark 1.** *The re-scaling constant $c'$ in the above theorem will not affect the value much. What truly matters in the function format is the parameter $c$, which affects how fast the weight function $g(\cdot)$ changes. Our value function choice picked $c$ as the natural number $e$.*

*Proof of Theorem 2.* The non-trivial part of the proof is to first prove $f(0) = 0$ and $g(0) = 1$, which are not clear in hindsight even given the two properties above. With these two "boundary" conditions, we will then be able to pin down the concrete format of $f$ and $g$.

Letting $a_S(D_{y_i}) \to 0$, we first have $\lim_{a_S(D_{y_i}) \to 0} f(a_S(D_{y_i}))g(0) = f(0)g(0) = 0$ which is at least $f(0)g(1)$ due to the Property 1. By monotonicity, we have for any $y \in [0, 1]$

$$0 = f(0)g(0) \le f(0)g(y) \le f(0)g(1) \le 0 \tag{10}$$

This implies that the inequalities above must all be tight, and thus $f(0)g(y) = 0$ for any $y$. Since $g(y)$ is not always 0, this implies $f(0) = 0$.

With $f(0) = 0$ as proven above, we are now ready to pin down the format of $g(\cdot)$. Then under the special case that $D_{y_i,1} = \emptyset$, we have $f(a_S(D_{y_i,1})) = f(0) = 0$ and thus the second property becomes

$$f(a_S(D_{y_i})) \cdot g(a_S(D_{-y_i})) = f(a_S(D_{y_i})) \cdot g(a_S(D_{-y_i,1})) \cdot g(a_S(D_{-y_i,2})) \tag{11}$$

for any $D_{-y_i} = D_{-y_i,1} \cup D_{-y_i,2}$. Plugging any $D_{y_i}$ such that $f(a_S(D_{y_i})) \neq 0$ into the above equality, we thus have

$$g(a_S(D_{-y_i})) = g(a_S(D_{-y_i,1})) \cdot g(a_S(D_{-y_i,2})).$$

Since $a_S(D_{-y_i})) = a_S(D_{-y_i,1}) + a_S(D_{-y_i,2})$, this implies $\log\big(g(a_S(D_{-y_i}))\big)$ is an additive function. That is, there exists $c''$ such that $\log\big(g(a_S(D_{-y_i}))\big) = c'' a_S(D_{-y_i})$, or equivalently, $g(a_S(D_{-y_i})) = e^{c'' a_S(D_{-y_i})} = c^{a_S(D_{-y_i})}$ for $c = e^{c''} > 1$.

Finally, we prove the format of $f(\cdot)$. The above proof for $g(\cdot)$ implies $g(a_S(\emptyset)) = c^{a_S(\emptyset)} = c^0 = 1$. Therefore, under the special case that $D_{-y_i} = \emptyset$, the second property becomes

$$f(a_S(D_{y_i})) = f(a_S(D_{y_i,1})) + f(a_S(D_{y_i,2})) \tag{12}$$

for any $D_{y_i} = D_{y_i,1} \cup D_{y_i,2}$. That is, $f$ must be an increasing linear function and thus there is a positive $c'$ such that $f(a_S(D_{y_i})) = c' \times a_S(D_{y_i})$. This concludes the proof of the theorem. $\qquad\square$

## 5 Experiments

### 5.1 Experiment setup

To compare with prior Shapley-based data valuation methods, we adopted most of the experiment setup from prior work, detailed along with other implementation details in Appendix A. In this section, we highlight some important details.

**Baseline methods.** We compare CS-SHAPLEY against three baselines: Data Shapley with the Truncated Monte Carlo approximation (TMC) [3], Beta Shapley [14], and Leave-One-Out (LOO) [2]. For Beta Shapley, we used the best $\alpha$ and $\beta$ values suggested in the original paper, which were also verified by our preliminary hyperparameter search. Note that another popular baseline method, KNN-Shapley [9], is also essentially covered by applying the data Shapley method to KNN classifiers.

**Evaluation tasks.** We adopted two benchmark evaluation tasks from prior work: high-value data removal and noisy label detection [3, 14]. In addition, we propose a new evaluation task to quantify

---

[3]We ignored the trivial situation that $c' = 0$ or $c = 1$, which is not interesting.

Table 1: Weighted accuracy drop for Logistic Regression and SVM-RBF using CS-SHAPLEY (CS), TMC-Shapley (TMC), Beta Shapley (Beta), and Leave-One-Out (LOO).

| Dataset | Logistic Regression | | | | SVM-RBF | | | |
|---|---|---|---|---|---|---|---|---|
| | CS | TMC | Beta | LOO | CS | TMC | Beta | LOO |
| CIFAR10 | **0.119** | 0.108 | 0.062 | 0.059 | **0.114** | 0.098 | 0.069 | 0.089 |
| Click | **0.053** | 0.007 | 0.017 | 0.016 | 0.004 | 0.004 | 0.004 | 0.004 |
| Covertype | **0.293** | 0.250 | 0.112 | 0.183 | 0.193 | **0.214** | 0.175 | 0.193 |
| CPU | 0.036 | 0.022 | 0.029 | **0.040** | **0.028** | 0.027 | 0.021 | 0.004 |
| Diabetes | **0.114** | 0.059 | 0.038 | 0.062 | **0.106** | 0.037 | 0.022 | -0.002 |
| FMNIST | **0.091** | 0.082 | 0.038 | 0.062 | **0.077** | 0.048 | 0.032 | 0.028 |
| MNIST-2 | **0.014** | 0.007 | 0.010 | 0.008 | 0.007 | 0.007 | 0.006 | 0.007 |
| MNIST-10 | **0.128** | 0.117 | 0.064 | 0.050 | 0.203 | **0.247** | 0.093 | 0.100 |
| Phoneme | **0.154** | 0.009 | 0.061 | 0.072 | **0.051** | 0.035 | 0.035 | 0.030 |

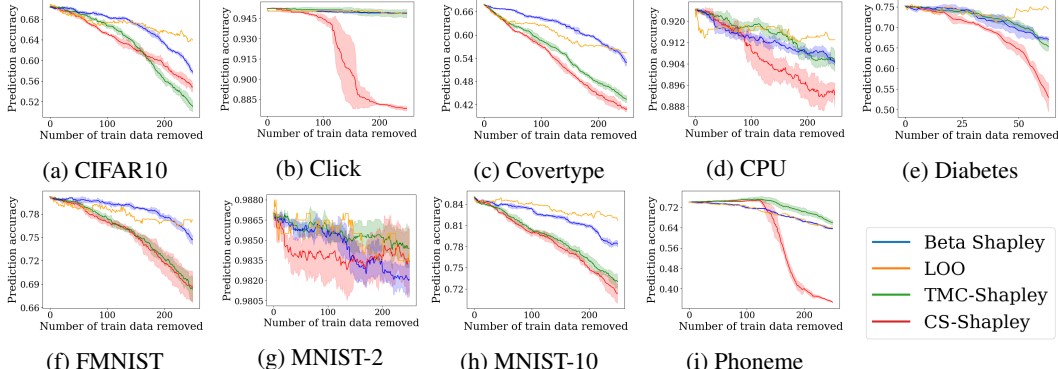

(a) CIFAR10    (b) Click    (c) Covertype    (d) CPU    (e) Diabetes

(f) FMNIST    (g) MNIST-2    (h) MNIST-10    (i) Phoneme

Figure 3: Performance across datasets when removing high-value instances for logistic regression.

the transferability of data value estimates across classifiers, to reveal a potential solution for mitigating the computational cost of estimating Shapley values for neural models.

**Datasets and classifiers.** We use nine benchmark datasets: Diabetes, CPU, Click, Covertype, CIFAR10 (binarized), FMNIST (binarized), MNIST (multi-class and binarized versions, denoted using -2 and -10, respectively), and Phoneme. When creating data subsets, we keep the original label distribution, instead of creating balanced subsets as in prior work. In addition, for each dataset and evaluation task, we systematically test the data valuation performance on four classifiers: logistic regression, SVM with the RBF kernel, KNN, and a gradient boosting classifier. We also include a multi-layer perceptron (MLP) as a target classifier to test the transferability of data values, since computing Shapley values with this classifier is prohibitively expensive.

**Summary of experiments in appendix:** Due to page limits, we report representative results in the main content and all additional results in Appendix C.

## 5.2 High-value data removal

Following the setup in prior work [3], for each valuation method, we gradually remove training instances from the highest value to the lowest value. At each removal step, we retrain the classifier and evaluate predictive performance on the held-out test data. Training instances with high value estimates should be helpful for model performance, so we measure the performance of each method with the accuracy drop following their removal. We follow prior work and plot the accuracy drop for up to 50% train data removed. To further quantify the performance differences observed in the plots, we also introduce a novel metric named *weighted accuracy drop*.

**Weighted Accuracy Drop.** An effective metric needs to evaluate two components underlying removal performance: 1) the total accuracy drop resulting from each valuation method, and 2) how quickly the drop in accuracy was achieved. Intuitively, the higher the relative value ranking of a data point, the more weight its impact on model performance should hold. We can therefore define the

Table 2: Area Under the Curve (AUC) for Logistic Regression and SVM-RBF using CS-SHAPLEY (CS), TMC-Shapley (TMC), Beta Shapley (Beta), and Leave-One-Out (LOO).

| Dataset | Logistic Regression | | | | SVM-RBF | | | |
|---------|------|------|------|------|------|------|------|------|
| | CS | TMC | Beta | LOO | CS | TMC | Beta | LOO |
| CIFAR10 | **0.450** | 0.429 | 0.424 | 0.275 | **0.387** | 0.317 | 0.321 | 0.272 |
| Click | **0.816** | 0.689 | 0.797 | 0.149 | **0.855** | 0.769 | 0.789 | 0.200 |
| Covertype | 0.706 | **0.766** | 0.653 | 0.179 | **0.712** | 0.618 | 0.600 | 0.196 |
| CPU | **0.785** | 0.779 | 0.654 | 0.207 | **0.808** | 0.671 | 0.516 | 0.189 |
| Diabetes | **0.441** | 0.355 | 0.435 | 0.194 | **0.412** | 0.362 | 0.400 | 0.210 |
| FMNIST | **0.570** | 0.554 | 0.552 | 0.340 | **0.512** | 0.382 | 0.412 | 0.239 |
| MNIST-2 | **0.831** | 0.815 | 0.806 | 0.280 | **0.837** | 0.663 | 0.611 | 0.300 |
| MNIST-10 | 0.877 | **0.933** | 0.845 | 0.371 | 0.674 | **0.747** | 0.510 | 0.254 |
| Phoneme | **0.575** | 0.535 | 0.416 | 0.222 | **0.579** | 0.555 | 0.496 | 0.255 |

*weighted accuracy drop* (WAD) as the summation of the cumulative accuracy drop at each removal step, weighed by the reciprocal of the removal step (i.e. reciprocal of the rank). Formally, for a training set $T = \{(x_i, y_i)\}_1^n$ sorted from the highest to the lowest value we have:

$$\text{WAD}_T = \sum_{j=1}^{n} \left( \frac{1}{j} \sum_{i=1}^{j} a_{T_{-\{1:i-1\}}}(D) - a_{T_{-\{1:i\}}}(D) \right)$$

where $T_{-\{1:i\}}$ represents the training set with the first $i$ instances removed based on the data valuation rank. When $i = 1$, $a_{T_{-\{1:i-1\}}}(D) = a_{T_{-\emptyset}}(D)$ equals the predictive accuracy with the full training set $T$. In effect, this enables us to assign high importance to the highest-ranked data points while still capturing the overall performance across removals, as depicted in the plots.

**Results.** We report the weighted accuracy drop using logistic regression and SVM with the RBF kernel across datasets in Table 1 and plot the removal performance of logistic regression in Figure 3. As shown, our method outperforms the baseline methods in most of the settings. Similar results are observed for the other two classifiers, as shown in Appendix C. This demonstrates the efficacy of using a value function that discriminates between in-class and out-of-class accuracy. For the SVM-RBF results on the Click dataset, we observe the identical performance across methods. Whereas prior work has used artificially balanced datasets, we performed stratified sampling to maintain the label distribution. In the case of Click, the dataset is highly imbalanced and SVM usually needs additional tricks to work on highly-imbalanced datasets [6].

### 5.3 Noisy label detection

To generate noisy training data, we shuffle the labels of a random 20% of the training data. We compute value estimates on the noised training sets using each valuation method and then simulate manual inspection by checking data labels from lowest value to highest value. The expectation is that an effective data valuation method will assign low values to mislabeled instances relative to the correctly labeled instances [3]. In our work, we use a rank-based approach to directly evaluate performance and visualize the retrieval results with a precision-recall (PR) curve. In addition, we also compute the Area Under the Curve (AUC) of the PR curve for quantitative results.

**Results.** We report AUC for logistic regression and SVM-RBF in Table 2. Similar to the removal experiments, our method has the best overall performance. We do note slightly weaker performance on the multi-class datasets compared to the removal experiments (see rows for Covertype and MNIST-10 in Table 1 and Table 2). This could be attributable to the simple sampling strategy of constructing $S_{-y_i}$. This suggests that in a multi-class setting, our method may benefit from increasing the minimum number of out-of-class samples.

### 5.4 Transferability of data values

Even with approximation, Shapley values can be computationally expensive to compute for larger models. For example, the experiments in subsection 5.2 had a 1:120 runtime ratio between the quickest (Diabetes) and longest (Covertype) running datasets on logistic regression. This would have

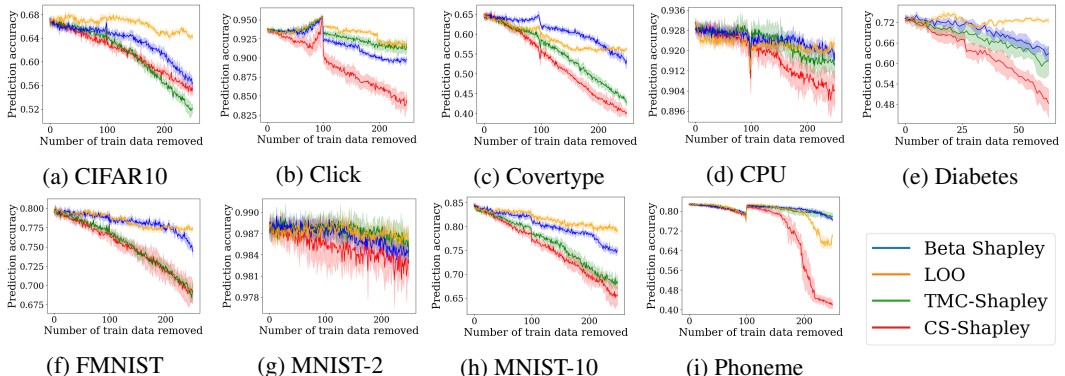

Figure 4: Performance across datasets when transferring values from logistic regression to MLP for the high-value removal task.

scaled to nearly 4-months to run MLP on Covertype.[4] It is therefore of great interest to understand to what extent Shapley values computed with a simple classifier can be transferred to other models, such as neural networks. In prior work, Jia et al. [11] demonstrated the efficacy of a specific case by using a KNN trained over pre-trained embeddings as a surrogate classifier for several target learning models. We generalize this idea and try to answer the question: *to what extent can Shapley-based data values computed with various simple classifiers be transferred and applied to other classifiers?*

To answer this question, we use each of the four classifiers in subsection 5.2 as the "source" classifiers and evaluate the computed data values with other "target" classifiers on the data removal task. In addition to the four classifiers, we also include an MLP classifier in this evaluation, for which the computational cost of TMC-Shapley was prohibitively large during our preliminary experiments. Specifically, for data values computed with a source classifier on a given dataset, at each removal timestep we remove an instance, retrain the target classifier, and evaluate predictive performance as in the original removal experiments. In this experiment, we would like to answer two questions: (1) is there a similar pattern of removal performance on the target classifiers as on the source classifiers; and (2) which source classifier and data valuation method causes the greatest performance drop on target classifiers, as this would indicate high applicability in a real world setting?

**Results.** Figure 4 shows transfer of logistic regression to MLP across all datasets, and we refer the reader to Figure 3 for the source removal plots. Our results suggest that in general, Shapley-based data values are transferable across classifiers. Specifically, across methods the overall pattern of performance drop from source to target classifier is closely aligned. While these results demonstrate that Shapley-based data value estimates are transferable from simpler models even to neural models, they also suggest that the valuation performance on the source classifier can be used as an indicator of how well the performance would be on a target classifier. As an implication of this, hyperparameter tuning to achieve better source performance may lead to even better transferability results. We leave this to future work. Additionally, this has implications for being able to gain the benefits of application (such as training data selection) for large neural networks. Further, this transferability may indicate that Shapley values capture some implicit data features that are generally beneficial or harmful to learning models. We leave it to future work to empirically test this. Finally, as a result of this transferability, we also observe that since our method outperformed other methods on the source classifier, CS-SHAPLEY also outperforms when transferred across classifiers, and overall, logistic regression is highly-effective as a source classifier.

## 6   Conclusion

In this work, we propose CS-SHAPLEY, a Shapley value with a new value function that discriminates between training instances' in-class and out-of-class contributions. Our theoretical analysis shows the proposed value function is (essentially) the unique function that satisfies two desirable properties for evaluating data values in classification. Further, our experiments demonstrate the effectiveness of CS-SHAPLEY over existing methods on high-value data removal, noisy label detection, and data

---

[4]See Appendix A for information on compute resources.

value transferability. Currently, the proposed method only works on classification problems. In future work, we will explore the possibility of extending the a similar idea to regression.

## Acknowledgements

This material is based upon work supported by the National Science Foundation under Grant No. SaTC 2124538 and an Amazon Research Award to Yangfeng Ji. We thank the anonymous reviewers for their helpful feedback and suggestions. We also thank Hanjie Chen and Wanyu Du, as well as the rest of the UVA Information and Language Processing Lab, for engaging discussions about this work.

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
