# OpenReview forum: "CS-Shapley: Class-wise Shapley Values for Data Valuation in Classification"
_NeurIPS.cc/2022/Conference — NeurIPS 2022 Accept_

### Official Review · Reviewer_47yJ · 2022-06-24

**Rating:** 5
**Confidence:** 5
**Soundness:** 3 good
**Presentation:** 3 good
**Contribution:** 3 good

**Summary:**

The authors studied the data valuation problem, pointing out that many existing methods are not able to differentiate in-class and out-of-class contributions (Figure 1). To address this issue, they proposed a new value function in equation (3) and the Class-wise Shapley (CS-SHAPLEY) in equation (4). They also discussed theoretical results, showing the new value function is a unique function (up to constant multiplication; Theorem 2) satisfying two properties. Experimental results on various evaluation tasks show that CS-SHAPLEY outperforms many existing data valuation methods.

**Questions:**

- Figure 1 presents the motivation for this work, but it needs implementation details.
    - how do you select two data points `i` and `j` in this experiment? Do you have a sense of why one sample increases class 1 accuracy while another point decreases?
    - The selection of S: is it based on one particular `S`? or an average of multiple `S`?

- If the CS-SHAPLEY is not a form of the Shapley value, could it mean that the Shapley value is suboptimal?

- (Minors)
    - One missing parenthesis in equation (1).
    - In equation 1, seemingly, the right-hand side is not dependent on $\mathcal{A}$.

**Limitations:**

As for the possible limitations, the authors discussed its inapplicability on regression in section 6. However, this is not a critical issue for this paper.

**Strengths And Weaknesses:**

- (+) An idea of differentiating in-class and out-of-class is conceptually reasonable and novel.
- (+) The paper is well-written: the main idea and experiments are clearly explained.
- (+) Extensive experiments have been conducted, showing the power of CS-SHAPLEY.

- (-) It is not clear how the theoretical results are connected to the usefulness of CS-SHAPLEY.
    - Why are the two properties desirable? It seems that the uniqueness of the value function is purely based on mathematical assumptions (continuity, monotonicity, the structure assumption in equation (7), and properties 1 and 2) that might not be directly essential in the data valuation problem. For instance, what is the meaning of equation (8) in terms of data valuation?
    - I believe the uniqueness of the value function (Theorem 2) is scientifically correct, but it is not about the CS-SHAPLEY but a value function. Given that CS-SHAPLEY is a specific function of the new value functions, I am wondering why this uniqueness theorem is important for CS-SHAPLEY.
- (-) The relationship between CS-SHAPLEY and the Shapley value: According to the definition of CS-SHAPLEY in equation (5), I am less sure if CS-SHAPLEY has a form of equation (1) with the Shapley value in (3).
    - Does CS-SHAPLEY satisfy the axioms in section 3.1 (e.g., Symmetry, Linearity, Null player, and Efficiency)? or is CS-SHAPLEY a type of the Shapley value for a certain cooperative game? If CS-SHAPLEY is not a form of the Shapley value, the use of `SHAPLEY` in `CS-SHAPLEY` might be confusing.
- (-) In all the current results in section 5, due to the lack of standard error information, it makes it difficult to how significantly better CS-SHAPLEY is than other methods?

---

> ### Author Response · Authors · 2022-08-02
> **Author Response to Official Review of Paper10976 by Reviewer 47yJ**
>
> We thank the reviewer for their time and thoughtful review. We believe the reviewer’s rating may be based on some misunderstanding of our approach and the experiments, and thus respectfully ask the reviewer to re-evaluate the merit of this paper based on the clarifications below.
>
> **Re: the relationship between the theoretical results and CS-Shapley**
>
> - **Re: importance of the uniqueness theorem:**
>
> 	- The essential idea of CS-Shapley is a Shapley value with a carefully designed value function, therefore, the theoretical analysis justifies the choice of the value function. The uniqueness theorem justifies our design by proving the value function is essentially unique for the purpose of better capturing the value of a data point to its in-class classification.
>
> - **Re: why the two properties are desirable:**
>
> 	- The two properties are the essence of the new valuation function and reflect our argument about (1) differentiating in-class accuracy and out-of-class accuracy; and (2) the role of in-class and out-of-class accuracies in evaluating data contribution.
>
> **Re: the relationship between CS-Shapley and the Shapley value**
>
> - CS-Shapley is one type of Shapley value, just with a carefully chosen value function. So it does satisfy all the axioms of the Shapley value.
> -   To be specific, if we substitute equation 4 into equation 5 and switch the order of the two summations, then we have that $\phi_i$ is calculated as
> $$\phi_i = \sum_{S_{y_i}\subseteq T_{y_i}\backslash\\{i\\}} \frac{1}{{n-1}\choose |S_{y_i}|} E[v_{y_i}(S_{y_i}\cup \\{i\\}|S_{-y_i})] - E[v_{y_i}(S_{y_i}|S_{-y_i})]$$
> where $E[v_{y_i}(S_{y_i}\cup \\{i\\}|S_{-y_i})] = \frac{1}{2^{|T_{-y_i}|}} \sum_{S_{-y_i}} v_{y_i}(S_{y_i}\cup \\{i\\}|S_{-y_i})$
>  - Therefore, $\phi_i$ defined in equation 5 is still a Shapley value of a cooperative game of in-class training examples, under all possible out-of-class environments $\{S_{-y_i}\}$.
>
> **Re: lack of standard error information**
>
> - There might be some misunderstanding here. All reported results reflect the mean of 5 trials, with standard error information for every experiment reported using the standard deviation (as in prior work [3]) which is depicted using shaded regions in each figure. The detailed implementation is described in Appendix A.3.
>
> **Re: implementation details for Figure 1**
>
> - We picked two particular examples for a specific S to demonstrate the two opposite ways a data point could contribute to the in-class and out-of-class accuracies. We follow the experiment setup described in section 5 to compute each data point’s marginal contribution to a particular subset S and identified two points where the overall accuracy change was equivalent but differed by how the point impacted each class.
> - In practice (as demonstrated in 5.3), data points that decrease in-class accuracy could be noisy or mislabeled. We will clarify this in the next version of the paper.

---

> > ### Comment · Reviewer_47yJ · 2022-08-05
> > **Follow-up questions**
> >
> > Thank the authors for their detailed responses. Some points have been addressed by the authors, but there are still some points that require further information. I hope the points will be discussed during the short discussion time.
> >
> > **About the lack of standard error information:** I meant that the standard error information is missing in Tables 1 and 2. For instance, in the MNIST-10 dataset, the weighted accuracy drop by CS is `0.128` and that by Data Shapley is `0.117`. Is the gap (0.011) significant?
> >
> > **(Additional minor points that are not related to the authors’ responses)**
> > - In Definition 1, the normalizing constant $\frac{1}{n}$ is missing.
> > - In equation (1), $ v( S \cup \set{ i } ) $ is used, but since $ S $ is a subset of $ T := \set{ (x_i, y_i) } _{i=1} ^n $, it is not appropriate to use $S \cup \set{ i }$ for an input for a function $ v $.
> > - Similarly, in equation (1) and line 94, $T \backslash \set{ i }$ is used but, it should be $\set{ 1, \dots, n } \backslash \set{ i }$.
> >
> > I posted the following question but the author's response addresses it, so I crossed it out.
> >
> > ~~**About the relationship and the Shapley value:** The authors claimed that the value function is carefully chosen and the CS-Shapley does satisfy all the axioms. However, it is not yet clear whether the authors’ claim is really true.~~
> >
> > ~~To make the question very clear, let me state the definition of the Shapley value (Definition 1) and the CS-Shapley (Authors’ response). For a set $i \in N := \set{ 1, \dots, n }$ and a value function $v : 2^{N} \to \mathbb{R}$, the (data) Shapley value of $i$-th data is defined as
> > \begin{align*}
> > 	\phi_i ^{\mathrm{data}} = \frac{1}{n} \sum_{S \subseteq N \backslash \set{ i } } \frac{ v(S\cup \set{ i } ) - v(S) }{\binom{n-1}{|S|}}.
> > \end{align*}
> > We now state the CS-Shapley: For a set $i \in N$, we first define a value function $v_{y_i} : 2^{N} \to \mathbb{R}$ as in Equation (3). The CS-SHAPLEY of $i$-th data is defined as
> > \begin{align*}
> > 	\phi_{i} ^{\mathrm{CS}} = \sum_{ S_{y_i} \subseteq N_{y_i} \backslash \set{ i } } \frac{ E[ v_{y_i} ( S_{y_i} \cup \set{ i } \mid S_{-y_i} )] - E[v_{y_i}( S_{y_i} \mid S_{-y_i} )] }{\binom{n-1}{|S_{y_i}|}},
> > \end{align*}
> > where $E[ v_{y_i}( S_{y_i} \cup \set{ i } \mid S_{-y_i} )] =  \frac{ 1 }{ 2^{ | N_{-y_i} | }}  \sum_{S_{-y_i}} v_{y_i}( S_{y_i} \cup \set{i} \mid S_{-y_i} )$.~~
> >
> > ~~As clearly shown by the two equations, the CS-Shapley is very different from the Shapley value in many aspects. Specifically,~~
> > - ~~The CS-Shapley is defined with `MULTIPLE` value functions $v_{y_i}$, while the Shapley value is defined with a value function. (In case of MNIST, there are 10 different value functions $v_{1}, \dots v_{10}$.)~~
> >   - ~~This is problematic because it is not clear for what value function the Efficiency holds. In the data Shapely, $\sum_{i=1} ^n \phi_i ^{\mathrm{data}} = v ( N )$. What is $\sum_{i=1} ^n \phi_i ^{\mathrm{CS}}$?~~
> >  - ~~Not sure if the summations are considered over the same set. Are $ \set{ S :S \subset N \backslash \set{ i } } $ and $\set{ S : S_{y_i} \subseteq N_{y_i} \backslash \set{ i } } $ same?~~
> >  - ~~The term $E[ v_{y_i} ( S_{y_i} \mid S_{-y_i} )]$ is not a function of a subset $S$, but a function of $S_{y_i} \cup N_{-y_i}$. It considers all possible `out-of-class environments`.~~
> >  - ~~The normalizing constants are different. ~~
> >
> > ~~These points confuse whether the CS-Shapley is a type of the Shapley value.~~

---

> > > ### Author Response · Authors · 2022-08-05
> > > **Author Response to Follow-Up Questions**
> > >
> > > We thank the reviewer for their further response and are glad we answered most of their questions!
> > >
> > > **Re: Standard Error Information:**
> > > * In addition to the standard error information currently included in the plots, for the next version of the paper we will include expanded tables in the Appendix that show the error information for the WAD and AUC metrics.
> > >
> > > **Re: Minor Points:**
> > > * As explained in prior work [3], the constant in the Data Shapley definition is arbitrary. In this work, we choose it to be 1, which is also the original definition of the Shapley value in [21].
> > > * About Equation 1, $S$ is actually a subset of $T \backslash \\{i\\}$ (see line 94), so we need $S \cup \\{i\\}$ to compute the contribution of data point $i$. As the reviewer may have also realized, we use the notations $T$ and $S$ here to represent the set of indices, following standard convention [3, 14, 21].
> > >
> > > Please let us know if there are any unresolved questions we can answer.

---

> > > > ### Comment · Reviewer_47yJ · 2022-08-05
> > > > **Thank you for the clarifications.**
> > > >
> > > > Thank you for the clarifications. The authors' responses clarified all the points.

---

### Official Review · Reviewer_DWEJ · 2022-06-28

**Rating:** 4
**Confidence:** 3
**Soundness:** 2 fair
**Presentation:** 3 good
**Contribution:** 2 fair

**Summary:**

This paper proposes Class-wise Shapley values as a new valuation method for data in classification tasks by designing a value function that _separately_ considers the in-class accuracy and out-of-class accuracies. The value function is shown to uniquely (up to the basis of the exponential function) satisfies two new properties concerning the value of data in classification tasks (w.r.t. in-class and out-of-class accuracies). The paper also provides experimental comparison with some existing data valuation methods on two prior evaluation tasks (high-value data removal and noisy label detection). In addition, the paper includes experimental observations about the transferrability (acorss classifiers) of Shapley-based data values.

**Questions:**

Clarification questions
- Is there a reference for using the term _development set_ or is there a reason why validation/hold-out/test set is not suitable? The latter options are more widely used and more receptive.
- The example give in Figure.1. Specifically, "Both points belong to class 1 and produce the same overall development accuracy change." How to see the _same_? Or what is the overall development accuracy change?
- In line 117, "we do not assign high value to instances that may be detrimental to the out-of-class performance". It seems to imply some other methods do this, is it true?
- In line 122, "For multi-class classification, $D_{y_i}$ has all the instances with labels other than $y_i$." Should it be $D_{-y_i}$?
- In line 142, "CS-Shapley value of instance $i$ _conditioned_ on any out-of-class environment $S_{-y_i}$ ..." How is this `conditioned' reflected in Equ.(4)? Or how is $S_{-y_i}$ used in Equ.(4)?
- In line 145, "environmental data $S_{-y} \subseteq T_{-y_i}$ ..." Should it be $S_{-y_i}$?
- In line 152, "Exactly computing $\phi_i$ in Equation (5) requires averaging over exponentially many $S_{-y_i}$,..." The exponentially many $S_{-y_i}$ is clear from Equ.(5). The question is, does Equ.(4) also involve exponentially many $S_{y_i}$? If so, does it mean that Equ.(5) somehow contains "nested" exponential complexities?
- In lines 236-237, "When creating data subsets, we keep the original label distribution, instead of creating balanced subsets as in prior work." It seems imbalanced data are not an issue, why? In particular, perhaps CS-Shapley can address this, but how about other baselines?
- About the introduced metric _Weighted Accuracy Drop_ (WAD)
    - Is there a comparison with the previous metrics to help understand the difference? In particular, it would be helpful to use this comparison to show WAD is a good choice.
    - The reciprocal of the rank is used as the weights. What about other monotonically decreasing functions?
- In lines 276-277, "CS-Shapley may benefit from increasing the minimum number of out-of-class samples?" What does "increasing the minimum number of out-of-class samples?" Does it mean increasing $S$ or $D$ and what are out-of-class samples?









**Limitations:**

The authors have mentioned this work is limited to classification and defer the extension to regression to future work.

**Strengths And Weaknesses:**

The writing of the paper is clear and concise and the theoretical derivations seem correct with suitably defined notations. The perspective of explictly considering in-class and out-of-class accuracies for data values in classification is new (to my knowledge) and is also compared with some intuitive/naive choices such as F1, precison or recall in Appendix. However, the motivation for separately considering in-class vs. out-of-class accuracies can be made clearer/stronger, i.e., _the why_. Following this, the motivation for the two additional properties (which essentially fix the value function) can be made clearer, too, i.e., _the how_. After proposing the CS-Shapley, which seems very computationally costly to calculate, it would be better to describe an approximation algorithm with theoretical guarantees so as to provide a sense for how to apply it and what error to expect in practice. For experiments, Sec. 5.4 presents empirical observations about Shapley-based data values in general, and its relevance specifically to CS-Shapley can be made more explicit.


Strengths
- The writing is clear and the theoretical derivations seem sound.
- The perspective of separately considering in-class and out-of-class accuracies is new.
- The overall presentaion is good in terms of organization and the authors included examples and plots to help explain their ideas.
- The experiments included several known baselines and are performed over various datasets and classifiers.

Weaknesses
- _The why_ The motivation for separately considering in-class vs. out-of-class accuracies is not so clear. In lines 111-113, "Along the previous lines of discussion, we suggest data for classification may contain implicit pre-existing coalitions based on class membership, which should be accounted for when evaluating contributions."
    - Speicfically, what are such implicit pre-existing coalitions were not so clear. Specific use-cases or practical application scenarios would help.
    - The motivation/reason for why such coalitions should be accounted for is not so clear. Empirical evidence or references that justify this would be useful.
- _The how_  The motivation for the specific way of separating in-class vs. out-of-class accuracies, namely Property 1. is not so clear. The authors argue "in-class accuracy should be prioritized." (line 181) In particular, in lines 182-183, "Concretely, the value function for getting positive in-class accuracy $a_S(D_{y_i})$ and $0$ out-of-class accuracy is no less than getting even perfect out-of-class accuracy but $0$ in-class accuracy." The justification for this statement can be made stronger. Should this statement be true _all the time_ for _all possible cases_?
- The practical applicability of CS-Shapley can be improved if paried with a corresponding approximation algorithm with theoretical guarantees. The cited previous works such as Data Shapley [3], Beta Shapley [14] and KNN Shapley [9] all paired their proposed/considered Shapley value with an approximation algorithm (including theoretical guarantees) which specifically exploited the defined value (e.g., gradient Shapley) and/or machine learning structural assumptions (e.g., truncation, KNN).
- The relevance for Sec. 5.4 (transferrability of data values) to CS-Shapley or to separately considering in-class vs. out-of-class accuracies is not so clear. It seems the results are general to Shapley-based data values, instead of CS-Shapley.

---

> ### Author Response · Authors · 2022-08-02
> **Author Response to Official Review of Paper10976 by Reviewer DWEJ**
>
> We thank the reviewer for their time and thoughtful review. We hope the following clarifies the reviewer’s concern and hopefully helps the reviewer to re-evaluate the merit of our paper.
>
> **Re: "The why":**
>
> - **Re: The motivation for separately considering in-class vs. out-of-class accuracies**
>
> 	- Our motivation primarily comes from the intuition, but we believe our method has been thoroughly justified by the empirical success in Section 5. We have also provided an intuitive real-world example in Figure 1 about why previous Shapley-based methods using the full development set accuracy as the value function may not be appropriate. This motivates our argument that data contribution may be better measured based on class membership.
>
> - **Re: "implicit pre-existing coalitions" meaning and "use-cases or practical application scenarios:"**
>
> 	- As explained in lines 111-113, we consider class membership as a potential implicit pre-existing coalitional structure of the data. The word “pre-existing” means the coalitions of training examples are defined based on their predefined labels.
> 	- We are not sure what “use-cases or practical application scenarios” the reviewer means, but our method has been extensively justified by the empirical success on benchmark datasets in Section 5. As CS-Shapley better quantifies the value of data in classification tasks, it is thus applicable to any scenario that motivates previous study of data valuation, particularly Shapley-based metrics.
>
> **Re: The motivation for the specific way of separating in-class vs. out-of-class accuracies (Property 1)**
>
> - Figure 1 provides a concrete example to illustrate why in-class accuracy should be prioritized. For lines 182-183,  an example "getting even perfect out-of-class accuracy but 0 in-class accuracy" is likely to be an adversary (e.g. from a data poisoning attack). A proper data valuation should not encourage adversarial examples like this.
> - In addition, we should not encourage the examples that help "getting even perfect out-of-class accuracy but 0 in-class accuracy",  because these examples could be outliers that significantly pull the decision boundary closer to themselves, which eventually hurts the performance on the development set.
> - Although we are not sure of the exact meaning of "all the time for all possible cases", we do think this statement should be true at least for the typical classification setup (e.g., learning to improve classification performance).
>
> **Re: the approximation and theoretical guarantees of CS-Shapley in comparison to prior work**
>
> - We believe the reviewer may have some misunderstanding here. Both the Data Shapley (TMC-Shapley and Gradient Shapley) [3] and Beta Shapley [14] papers use the standard Monte Carlo sampling method as an approximation, which is also the method we use in this paper. Our work, as well as previous papers [3, 14], relies on the theoretical guarantee presented in well-known computational game theory literature (see citations [17, 1] as mentioned in our paper’s lines 155-157). In the case of [3], further justification of truncation and Gradient Shapley is provided empirically.
> - KNN Shapley [9] does come with a more efficient algorithm with theoretical guarantees. However, the method in [9] heavily relies on the structure of the KNN approach. This approach does not appear applicable to general classification methods as studied in our paper nor in other papers like Data Shapley and Beta Shapley.
> - Finally, the choice of our proposed value function, which is also the main contribution of this work, is theoretically justified (Section 4).
>
> **Re: the transferability task in section 5.4 being a general task**
>
> - Indeed, this evaluation task (as well as the other two) are all about general data valuation. Otherwise, we may run into the risk of designing an evaluation that only favors our method.
> - We motivate the necessity of this task by the current computational infeasibility of performing Shapley-based data valuation for neural models. We believe this task demonstrates the possibility of using Shapley-based data valuation on computationally expensive models (see lines 279-287).
>
> We will have a follow-up post to address some specific clarification questions.

---

> > ### Author Response · Authors · 2022-08-02
> > **Author Response to Clarification Questions in Official Review of Paper10976 by Reviewer DWEJ**
> >
> > **Re: “Is there a reference for using the term development set or is there a reason why validation/hold-out/test set is not suitable?”**
> >
> > - As pointed out by some textbooks (e.g., Murphy’s Probabilistic Machine Learning: An Introduction), “validation set” and “development set” refer to the same set. Here, using “development set” is mostly a personal preference.
> >
> > **Re: Figure 1: “ How to see the same? Or what is the overall development accuracy change?”**
> >
> > - For the subset $S$, the overall development accuracy change is defined by the numerator of Equation 1 $(v(S \cup \\{i\\}) - v(S))$. $v(S)$ and $v(S \cup \\{i\\})$ are defined as the sum of the in-class and out-of-class accuracies (see Eq. 2, lines 126-127). In Figure 1, the overall development accuracy change when adding $i$ or $j$ is +0.0015.
> >
> > **Re: line 117 (potential for assigning high value to harmful instances)**
> >
> > - As explained in lines 28-31, all prior methods assign values based on the overall performance improvement, without differentiating between out-of-class and in-class performance. Please see the response to "Re: The motivation for separately considering in-class vs. out-of-class accuracies" for further discussion.
> >
> > **Re: line 142 (‘conditioned’ and $S_{-y_i}$ in Equation 4)**
> >
> > - $S_{-y_i}$ is used in two value functions of the numerator of Equation 4. The “conditioned” means the data value in Equation 4 is computed with a specific $S_{-y_i}$ (aka, the out-of-class environment)
> >
> > **Re: line 152 (computational complexity)**
> >
> > - The computational complexity is _not_ nested, it is $O(2^{|T_{y_i}|}\cdot 2^{|T_{-y_i}|}$, which is actually smaller than the original computational complexity of Data Shapley in [3], which is $O(2^{|T|}$).
> >
> > **Re: lines 236-237 (using imbalanced datasets)**
> >
> > - Theoretically, Shapley-based data valuation, by considering all possible coalitions, can avoid any potential issues caused by data imbalance. Although, the actual value estimation is impacted by the approximation methods.
> > -   However, other data valuation methods (e.g., LOO) may not share this merit.
> >
> > **Re: comparison to ‘previous metrics’**
> >
> > - As noted in line 249, prior works only evaluate the removal task using the plotted accuracy drop, so no quantitative metrics have been previously used. This motivated our development of WAD to evaluate this common task. We include the standard plots in addition to WAD as a comparison and to demonstrate the relationship between the qualitative observations of the prior evaluation method and the quantitative measure of WAD.
> >
> > **Re: lines 276-277 (meaning of ‘increasing minimum number of out-of-class samples’)**
> >
> > -   Out-of-class samples refers to $S_-y_i,$ which is a randomly sampled subset of training instances that do not belong to the class $y_i$ (we provide pseudocode for our algorithm in Appendix A.1).
> > -   This is a comment for a practical issue. Theoretically, $S_{-y_i}$ can be any size. But in practice, we need to make sure $S_{-y_i}$ contains a certain number of examples to represent the out-of-class distribution.
> > -   For example, in multi-class settings (our multi-class datasets each had 10 classes), to provide a better estimate of the performance for each of the 9 classes not in $y_i$, our method may benefit from increasing the minimum instances when sampling $S_{-y_i}$ so the model doesn’t overfit on a small number of training instances per class.

---

> > > ### Comment · Reviewer_DWEJ · 2022-08-08
> > > **Post-rebuttal response**
> > >
> > > I thank the authors for the detailed response to clarify my questions, and wish to point out some unaddressed points and some reservation(s) that remain.
> > >
> > > ### Questions:
> > > 1\. In my original review:
> > > > In line 145, "environmental data $S_{-y}\subseteq T_{-y_i}$ ..." Should it be $S_{-y_i}$?
> > >
> > > was not directly addressed?
> > >
> > > 2\. Regarding the computational complexity, I see that the complexity is $O(2^{|T_{y_i}|} \cdot 2^{|T_{-y_i}|})$. My follow-up question is that, since $T_{y_i},T_{-y_i}$ form a partition of $T$, why is it
> > > >  ... actually smaller than the original computational complexity
> > > of Data Shapley in [3], which is $O(2^{|T|})$ )
> > >
> > > My understanding is that $|T_{y_i}| + |T_{-y_i}| = |T|$.
> > >
> > > 3\. Regarding "pre-existing coalitions".
> > > > As explained in lines 111-113, we consider class membership as a potential implicit pre-existing coalitional structure of the data. The word “pre-existing” means the coalitions of training examples are defined based on their predefined labels.
> > >
> > > Are the coalitions w.r.t. data points, as in some data points form a certain coalition? As the term _coalition_ is used in cooperative/coalitional game theory and the Shapley value to describe a subset of players which in this case seem to be the data points.
> > >
> > > If so, my question is: since _coalition structure_ is used in coalitional game theory as a solution to a game instead of a characterization of the players, is the phrase pre-existing coalition structure meant to be a solution as well?
> > >
> > > Furthermore, could the authors provide one specific example of such potential implicit pre-existing coalitional structure or alternatively provide a mathematical formalization to improve the clarity of what it represents or how it is formed? Perhaps in the revision since the discussion period is relatively short.

---

> > > > ### Comment · Reviewer_DWEJ · 2022-08-08
> > > > **Post-rebuttal response (continued)**
> > > >
> > > >
> > > > ### Reservation and explanation:
> > > > My main reservation is that despite the new perspective of separating in-class and out-of-class accuracies (pointed out as one of the strengths in my review), the motivation for it and the specific formulation can be further improved. Specifically, see below:
> > > >
> > > > 1\. The motivation for the problem is
> > > > > Our motivation primarily comes from the intuition, but we believe our method has been thoroughly justified by the empirical success in Section 5.
> > > >
> > > > While I believe using an intuition to inspire a motivation is acceptable, using only the intuition as the entire motivation is not very strong. In particular, the authors provide a concrete but artificial example (Figure 1) to illustrate this problem. In my opinion, if such problem (which clearly requires separate consideration for in-class vs. out-of-class accuracies) is more prominently observed in specific scenarios (theoretical and/or empirical), the motivation would be stronger.
> > > >
> > > > 2\. The motivation for the solution of separating the in-class and out-of-class accuracies as in Equ.(3) includes (from author's response)
> > > > > For lines 182-183, an example "getting even perfect out-of-class accuracy but 0 in-class accuracy" is likely to be an adversary (e.g. from a data poisoning attack). A proper data valuation should not encourage adversarial examples like this.
> > > >
> > > > > because these examples could be outliers that significantly pull the decision boundary closer to themselves, which eventually hurts the performance on the development set.
> > > >
> > > > For simplicity, denote the first bullet point as _adversary_, the second bullet point as _decision boundary_. Considerations from adversary and decision boundary do not seem to be included in the introduction or Sec. 3.2 to motivate the specific formula in Equ. (3). In addition, w.r.t. decision boundary, while I concede the statement makes intuitive sense, without (theoretical or empirical) evidence, it is difficult to be very convincing.
> > > >
> > > > Lastly,
> > > > > Although we are not sure of the exact meaning of "all the time for all possible cases", we do think this statement should be true at least for the typical classification setup (e.g., learning to improve classification performance).
> > > >
> > > > Let me clarify "all the time for all possibles" by framing it as a question: when do the authors believe that the statement should not be true?
> > > >
> > > > To put things together, my reservation revolves around the motivation behind the problem (in terms of when and where it exists) and the solution (in terms of formal treatments). The question I raised about whether the statement in lines 182-183 (about Property 1) is meant to understand the limitations of such property and thus the proposed valuation method. Based on this, I thus believe the experiments would also benefit from a clearer exposition into the problem and solution. Specifically, by clearly identifying the problem (and when and where it exists), experiments can be designed to specifically evaluate the performance of solutions on this problem. In contrast, the current experiments use many existing evaluation metrics (though WAD is a new and more general formula but not specific to the considered class-wise valuation). Note that this is not to discredit the existing evaluation metrics, but I believe specific ones should be considered to showcase the effectiveness of the proposed solution to this problem. To ease a possible concern of the authors: "Otherwise, we may run into the risk of designing an evaluation that only favors our method", I believe that as long as the evaluation includes previous well-known metrics (i.e., not only the specific/custom metrics), then this issue would not be as significant since that is what using those previous well-known metrics is for. Similarly, the motivation for experiment on transferrability is not so clear since it (as the authors state) "are all about general data valuation" so it is not specific to this problem or solution.

---

> > > > > ### Author Response · Authors · 2022-08-09
> > > > > **Author Response**
> > > > >
> > > > > **Re (1): empirical and theoretical evidence for motivation**
> > > > > * The example in Figure 1 is not artificial. As explained in the caption, it is a real example from the CIFAR10 dataset, used to provide empirical evidence that full development set accuracy can obscure points that may harm their own classes.
> > > > > * We note that our use of motivating intuition is common in prior work. For example, in [3] the authors motivate Data Shapley using a simple, intuitive example of why leave-one-out may fail on a nearest-neighbor classifier.
> > > > > * In addition to the empirical evidence in Figure 1, we also consider the extensive experiment results as empirical evidence, which also shows the necessity of separate consideration for in-class and out-of-class accuracies.
> > > > > * About providing theoretical evidence, as far as we know,
> > > > >    * The existing work in the line of Data Shapley [3] considers classification as a cooperative game with some examples.
> > > > >    * On the other hand, [Couellan, 2017] formulates SVM classification as a non-cooperative game with some geometric intuitions.
> > > > >    * A theoretical justification in this case would be equivalent to “what kind of game a classification problem should be?“, which we think is still an open question based on our understanding of prior work (and are happy to know if we missed some critical prior work).
> > > > > * Nevertheless, it’s good to know that our explanations intuitively make sense. We expect this work can stimulate some future work on theoretical analysis.
> > > > >
> > > > > [Couellan, 2017] Couellan. A note on supervised classification and Nash-equilibrium problems. 2017
> > > > >
> > > > > **Re (2): motivation of specific way of separating in-class and out-of-class accuracies**
> > > > > * About the impact of outliers on decision boundary: there are some classical works on how outliers could impact the decision boundary of a classifier, for example (Xu, et al., 2006).
> > > > > * In lines 41-42 we mention consideration for adversaries. In many cases, adversarial examples are considered as some types of outliers, e.g., (Grosse, et al., 2017; Manohar-Alers et al., 2021).
> > > > > * Since one is a special case of the other, our statement in general is consistent. We also note we provided this as an additional intuitive example for the purpose of illustration in this discussion, and are glad it makes intuitive sense!
> > > > > * We cite our previous response: we do think this statement should be true at least for the typical classification setup (e.g., learning to improve classification performance), so we don’t know of scenarios where it should not be true.
> > > > >
> > > > > (Xu et al., 2006): https://www.aaai.org/Papers/AAAI/2006/AAAI06-086.pdf
> > > > >
> > > > > (Grosse et al., 2017): https://arxiv.org/pdf/1702.06280.pdf
> > > > >
> > > > > (Manohar-Alers et al., 2021): https://openreview.net/pdf?id=XDo0go2IJgT
> > > > >
> > > > > **Re: evaluation design**
> > > > > * Thank you for the clarification. If our understanding is correct, the reviewer may expect either 1) some specific evaluation strategies that can demonstrate the value of separating in-class and out-of-class accuracies, or 2) some specific evaluation strategies that can demonstrate the value of the specific way of separating in-class and out-of-class accuracies.
> > > > > * For the specific evaluation of 1, two possible things that we can do are:
> > > > >    * Compare the data values between TMC-Shapley (or other prior work) and CS-Shapley for one specific class
> > > > >    * Compare the data values between CS-Shapley and the ones only using in-class accuracy
> > > > > * Alternatively, for the specific evaluation of 2, two possible things that we can do that use alterations that violate Property 1 are:
> > > > >    * Compare the data values with the proposed value function and an alteration where $g$ is monotonically decreasing, i.e. $g=e^{-a_S(D_{-y_i})}$
> > > > >    * Compare the data values with the proposed value function and an alteration where out-of-class accuracy is prioritized (i.e. $v_{y_i}=a_s(D_{-y_i}) \cdot e^{a_S(D_{y_i})}$)
> > > > > * In any case, both experiments should not be difficult to conduct and we can add them in the paper revision.
> > > > > * While the transferability experiment is motivated similarly to the existing evaluation tasks (i.e. compare performance of different methods on potential data valuation application scenarios), we hope the proposed additional evaluation will address the reviewer’s concerns.
> > > > >
> > > > > Please let us know whether our understanding is correct and any comments on the proposed evaluation strategies.

---

> > > > ### Author Response · Authors · 2022-08-09
> > > > **Author Response**
> > > >
> > > > **Re: typo**
> > > > * We addressed this in the general comment “Rebuttal Revision” and fixed it in the rebuttal draft that we previously uploaded. We appreciate you finding the notation typo!
> > > >
> > > > **Re: computational complexity**
> > > > * Using the big-O notation may not be clear enough, so we would like to offer a specific toy example instead. Consider the set with label $y_i$ that has two examples, and the set with $-y_i$ that also has two examples. As the computation of CS-Shapley cannot operate on the empty set, we have $3$ non-empty subsets from $y_i$ and $3$ non-empty subsets with $-y_i$. In total, they give $9 (=3 \times 3)$ combinations. While with Data Shapley, the number of all possible subsets is $15 (=2^4 - 1)$.
> > > >
> > > > **Re: coalitions**
> > > > * The data points form the coalitions, as is consistent across Shapley-based data valuation methods [3, 14].
> > > > * The phrase “pre-existing coalitional structure” is not meant to be a solution. In coalitional game theory, a solution is typically some way to “fairly” allocate the total rewards. Shapley value is one possible solution concept (it became a quite popular one due to its axiomatic interpretation and uniqueness). Other popular solution concepts include, e.g., the “core” or “nucleolus”, which satisfy slightly different properties than the Shapley value. It is possible that we may have misunderstood the reviewer’s comment, but these solution concepts are not “coalition structure” but rather are allocation functions based on rewards generated by some coalition structure, where a coalition structure is a partition of the set of players (i.e. data points in a data valuation setting) [Aumann & Dreze, 1974].
> > > > * In this work, we consider class membership as a potential pre-existing coalition structure in the data. In other words, our proposed method considers class membership as a pre-existing partition in the data that should be accounted for in classification problems.
> > > >
> > > > Aumann, R. J., & Dreze, J. H. (1974). Cooperative games with coalition structures. International Journal of game theory, 3(4), 217-237.

---

### Official Review · Reviewer_kNhs · 2022-07-11

**Rating:** 6
**Confidence:** 2
**Soundness:** 3 good
**Presentation:** 3 good
**Contribution:** 3 good

**Summary:**


Class-wise Shapley Values (CS-SHAPLEY) is a data valuation algorithm for binary and multiclass classification algorithms.

CS-SHAPLEY distinguishes between training instances’ in-class and out-of-class contributions. Then, the algorithm combines the 2 measurements into one value.

CS-SHAPLEY is the unique function that satisfies 2 properties for evaluating data values in classification:
Property 1 (Priority of In-class Accuracy) makes train points that improve their in-class accuracy more important than those that improve the out-of-class accuracy.
Property 2 (In-class Value Additivity and Out-of-class Weight Discounting) makes the data values and weights additive.

CS-SHAPLEY takes too long to compute. Its approximation is compared against Truncated Monte-Carlo (TMC), Beta Shapley, and Leave-One-Out and did perform better in most of the baseline tasks.


**Questions:**

Figure 3: how do you explain that other methods like LOO are more efficient at the beginning (first 20 train points removed)? Have you thought about an ensemble method?


Concerning the sub-plot "(e) Diabetes," why is it limited to less than 75 data points?


# Relation To Prior Work
Authors did consider popular contributions like Data-Shapley, LOO, and Beta Shapley.
Did you consider G-Shapley? (see [3])

# Additional Feedback

Did you consider the effect of hyperparameter tuning of the ML algorithm on the data valuation? i.e., the minimum starting score of the considered tasks is 0.68. Would CS-Shapley be the best if the starting accuracy was 0.3?

**Limitations:**

no negative societal impact

**Strengths And Weaknesses:**

# Strengths

CS-SHAPLEY is an original approach to valuate train data points. It is the only one distinguishing between in-class and out-of-class accuracy.
The authors propose a method that is analyzed using both axioms/properties and practical evaluation tasks (benchmark).
CS-SHAPLEY performs well on various tabular data.

Authors provide a promising metric to objectively evaluate data valuation algorithms on the task of removing high-value instances. Rather than visually comparing curves, the authors propose a one value score that considers the order in which data points were removed. Nevertheless, this metric seems very close to the AUC. A small comparison between both would be interesting.


# Weaknesses

High-value data removal might not be sufficient to confirm that the proposed method is performing better: As explained by the authors, data removal might cause imbalance and a faster drop in performance. Therefore, it would be better to complement the experiment with the opposite task (adding data points)



The authors did not reuse the same datasets and tasks used in prior work. Therefore it is still difficult to confirm that this method is better. Please include, for example, the Skin Cancer dataset used in the TMC-Shapley paper [3].

# Correctness

Random data removal is a good baseline to check if, eventually, all algorithms are failing a specific task. Please include it to confirm that CS-SHAPLEY is better.

# Clarity

Line 92: the definition of v(S) and a_S(D) are condensed into one sentence, which makes it ambiguous.

Figure 3: adding an x-axis with the percentage of data removed would help better understand the real contribution of these deleted data points. a more straightforward fix is to add the total train set size to the x-axis title
Please add to the y-axis an arrow down to express that "lower is better".

The abbreviation TMC was defined but never used. Instead, the authors did use "Data-Shapley". Please use TMC-Shapley as proposed in the original paper hoping to make it clearer for the readers.



# Additional Feedback

Figure 3: you could remove the redundant y-axis's title.

---

> ### Author Response · Authors · 2022-08-02
> **Author Response to Official Review of Paper10976 by Reviewer kNhs**
>
> We thank the reviewer for their thoughtful review and suggestions. We believe some weaknesses in the comments are due to some misunderstandings of the evaluation methods.
>
> **Re: removing vs. adding high-value data points**
>
> - Prior work (e.g., Figure 1 in [3]) shows the overall ranking of valuation methods when adding high-value data points is consistent with the ranking of valuation methods when removing high-value points, which was also observed in our preliminary results.
> - Therefore, we chose to save some space for other evaluation methods that can privide complementary information.
>
> **Re: using evaluation tasks and datasets from prior work**
>
> - There may be some misunderstanding. Two of our evaluation tasks (data removal and label noise detection) are adapted directly using the experimental setup of prior work [3, 14].
> - Also, we use the following datasets from prior work: FMNIST from [3]; Covertype, CIFAR10, FMNIST, MNIST, Click, Phoneme, and CPU from [14].
>
> **Re: random data removal baseline**
>
> - In all the prior works, the random baseline is already consistently outperformed by all methods, which is why we did not use it as a baseline. Nevertheless, as confirmation, during the rebuttal period we ran logistic regression on the Diabetes dataset using the random baseline. Consistent with the results in prior work, the random baseline was outperformed by all methods (WAD = 0.019, see Table 1 for comparison).
>
> **Re: LOO efficiency on first 20 data points in Figure 3**
>
> - We consider it dependent on the specific data distribution and classifier, for example, this is observed on the CPU dataset with logistic regression in Figure 3d, but _not_ for SVM in Figure 5d (for additional comparisons, see Figure {3, 5-7}).
>
> **Re: number of data points in Diabetes subplot**
>
> - As explained in Appendix A.2 (Data Usage), our dataset split ratio (1:1:4) is consistent with prior work. Since the Diabetes dataset is only 768 instances, we use a 128/128/512 train/dev/test split and the visualization results only show 50% of the training set (see lines 249-250).
>
> **Re: considering G-Shapley**
>
> - As the original work ([3]) shows that TMC-Shapley consistently outperforms G-Shapley on benchmark datasets, we decided to not include G-Shapley.
>
> **Re: hyperparameter tuning of the ML algorithm**
>
> - For this work, we selected the best performing hyperparameters for each model. Our understanding is that a poorly tuned model may have lower stability when performing Shapley-based data valuation, which may decrease the efficacy of any method.
> - The hyperparameter tuning is described in Appendix A.3.
>
> We hope our responses have addressed the reviewer’s concerns. We are happy to answer any further questions!

---

> > ### Comment · Reviewer_kNhs · 2022-08-05
> > **all clear**
> >
> > Thanks for the clarification.

---

### Official Review · Reviewer_HSku · 2022-07-11

**Rating:** 7
**Confidence:** 4
**Soundness:** 3 good
**Presentation:** 3 good
**Contribution:** 3 good

**Summary:**

The paper investigates the problem of data valuation using Shapley value-based approximation methods. The authors propose a new value function that incorporates the notion of in-class and out-of-class accuracy as opposed to the baseline approach of using overall accuracy. This approach is justified theoretically and shows improvement in empirical evaluation. The authors also investigate how well data valuation methods transfer across classifiers.

**Questions:**

* Is it possible to use other metrics (e.g. AUC) for $a_S(D_{y_i})$ and $a_S(D_{-y_i})$?
* In Figure 4, it looks like a number of datasets have a sharp increase or decrease around where 100 samples are removed. Do you have any thoughts as to why this occurs?

**Limitations:**

Limitations are adequately discussed in Section 6.

**Strengths And Weaknesses:**

**Strengths:**
* The proposed value function is a novel approach that improves performance over existing methods. Comparisons are made to 3 competing data valuation methods over 8 different datasets.
* The proposed method is well-motivated and intuitive. The theoretical justifications are sound.

**Weaknesses:**
* The proposed method is limited to classification tasks, and only the accuracy metric is investigated for the value function.
* No sensitivity analysis or justification of choice in c, c' parameters. Adding this analysis would improve the paper.


**Miscellaneous Minor Issues / Suggestions**
* Eq. 1 is missing a right parenthesis: $v(S \cup \{ i \})$"
* Line 122 "For multi-class classification, $D_{y_i}$ has all the instances with labels other than $y_i$." I think this should be $D_{-y_i}$?
* Theorem 2 has an extra right parenthesis in $v_{y_i}(S)$
* It would be helpful if the shaded area in the plots (Figure 3,4) representing standard deviation was defined in the table caption.
* Error bars / confidence intervals in the table results would be helpful as a number of the results are very close.
* It looks like setting $c=1$ and $c' = 1$ would recover the value function for the standard Shapley value. It might be interesting to see whether you can interpolate between the results of standard and classwise Shapley values depending on choice of $c, c'$.

**Summary:**
In general, the proposed method is a novel and intuitive approach which is justified both theoretically and empirically. The paper is also well-written and well-motivated. There are a some areas for improvement, such as extending to metrics other than accuracy and additional sensitivity analysis, however the paper in its current form seems sufficient for acceptance.

---

> ### Author Response · Authors · 2022-08-02
> **Author Response to Official Review of Paper10976 by Reviewer HSku**
>
> We thank the reviewer for appreciating our work. In the following, we briefly address the reviewer's two concerns.
>
> **Re: limited to classification tasks**
>
> - Indeed, as the first method of Shapley value that differentiates data contribution to different values of output ($y$) , our design is for classification tasks and takes advantage of discrete output values.
> - Extending a similar idea to regression cases is quite intriguing but also adds an additional layer of challenge (e.g. how to define a measurement like Eq. 2 in regression tasks), which will be explored in future work.
>
> **Re: using accuracy and other metrics (e.g. AUC)**
>
> - A major reason for choosing accuracy as the metric is to compare the class-wise distinction with prior work, as all prior works use accuracy as their value function. However, we believe all methods (including ours) can be applied to other metrics without the need of much modification.
> -  It is possible to use other metrics. For example, if it is the AUC of the PR curve, then the only thing we need to do is to redefine the Precision and Recall using equations similar to Eq. 2
>
> **Re: sensitivity analysis of $c$ and $c'$**
>
> -  The choice of $c’$, as a coefficient, does not impact the results. In addition, the value of $c$ won’t change the ranking of data values, so it will also not impact the evaluation results in this work.
> - Nevertheless, we appreciate the reviewer's thoughtful suggestion, and will add a discussion about the choice of $c$ and $c'$ in the final version.
>
>
> **Re: the sharp increase or decrease**
>
> - Interestingly, this observation at 20% data removal occurs on several datasets only when transferring to MLP (see Figures 16, 20, and 24 for additional MLP transfer results). Therefore, we suspect it is due to the specific target classifier. However, we do not yet have theoretical justification for this observation.

---

> > ### Comment · Reviewer_HSku · 2022-08-05
> > **Rebuttal Response**
> >
> > Thank you for your responses. I appreciate you adding the discussion regarding c and c'.

---

### Author Response · Authors · 2022-08-02
**Rebuttal Revision**

The current revision reflects corrections of several notation typos and minor alterations made based on the suggestions of the reviewers. Revisions are reflected in red. Due to current page limits, additional suggestions will be incorporated into the final paper.

---

### Author Response · Authors · 2022-08-09
**Thank you for the comments and suggestions**

We thank the reviewers for the helpful comments and suggestions, and are glad we were able to provide clarifications and address all of the questions. If there are any followup questions, we will be happy to answer them.

---

### Meta-Review · Area_Chair_HTLg · 2022-08-23

**Recommendation:** Accept
**Confidence:** Certain

**Metareview:**

The reviewers agree that the paper makes a potentially interesting contribution and clearly holds promise, but also that it needs another round of revision. The authors' rebuttal was very much appreciated and could indeed clarify some misunderstandings (e.g., a single vs. several cooperative games), though it also left some questions open (and even gave rise to a few others). Eventually, the reviewers' reservations regarding the proper formalisation of the problem, the meaning and interpretation of the approach (e.g., meaning of comparing many Shapley values from different cooperative games), as well as the experimental validation couldn't be completely resolved.

**Award:**

No

---

### Decision · Program_Chairs · 2022-09-14

Accept